# Comparing Luojia 1-01 and VIIRS Nighttime Light Data in Detecting Urban Spatial Structure Using a Threshold-Based Kernel Density Estimation

Yuping Wang and Zehao Shen *

Ministry of Education Key Laboratory for Earth Surface Processes, College of Urban and Environmental Sciences, Peking University, Beijing 100871, China; yupinwang@pku.edu.cn
* Correspondence: shzh@urban.pku.edu.cn; Tel.: +86-010-62751179

**Abstract:** Nighttime light (NTL) data are increasingly used in urban studies and urban planning owing to their strong connection with human activities, although the detection capacity is limited by the spatial resolution of older data. In the present study, we comparedthe results of extractions of urban built-up areas using data obtained from the first professional NTL satellite Luojia 1-01 with a resolution of 130 m and the Visible Infrared Imaging Radiometer Suite (VIIRS). We applied an analyzing framework combing kernel density estimation (KDE) under different search radii and threshold-based extraction to detect the boundary and spatial structure of urban areas. The results showed that: (1) Benefiting from a higher spatial resolution, Luojia 1-01 data was more sensitive in detecting new emerging urban built-up areas, thus better reflected the spatial structure of urban system, and can achieve a higher extraction accuracy than that of VIIRS data; (2) Combining with a proper threshold, KDE improves the extraction accuracy of NTL data by making use of the spatial autocorrelation of nighttime light, thus better detects the scale of the spatial pattern of urban built-up areas; (3) A proper searching radius for KDE is critical for achieving the optimal result, which was 1000 m for Luojia 1-01 and 1600 m for VIIRS in this study. Our findings indicate the usefulness of the KDE method in applying the upcoming high-resolution NTL data such as Luojia 1-01 data in urban spatial analysis and planning.

**Keywords:** kernel density estimation; Luojia 1-01 satellite; nighttime light; spatial resolution; searching radius threshold; urban built-up area

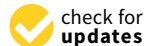



## 1. Introduction

Cities comprise a landscape type with the most concentrated human activities. The intense exchanges of materials, energy, and information connect cities with nearby areas via traffic and other networks, and form apparent social and environment gradients from urban to rural areas, with diverse city structures [1]. Spatial analysis of city structures is critical for understanding city functions and evolution, while accurate discrimination ofthe urban boundary and the internal structure is a prerequisite for further spatial analyses [2–4]. In general, urban built-up areas within the administrative region of a specific city comprise continuous areas with adequate municipal facilities [5]. Urban built-up areas are the core areas of cities and the main focus of research into urban structure, functioning, anddevelopment [6].

Remote sensing images are a major data source for urban structure analysis. Early studies usually employed daytime remote sensing data such as Thematic Mapper (TM) images to extract urban built-up areas [7,8], but information about the buildings is not always an accurate indicator of theintensity and economic importance of urban areas. In recent years, nighttime light (NTL) data have been increasingly employed to indicate human activities at the landscape to regional scales [9]. Pioneered by Elvidge et al.'s application of NTL data in city mapping and analyses [10,11], this new approach provided

a distinct and sometimes more effective informationfor urban structure identification and spatial analysis, especially for high energy release patterns [11,12].

Li and Li (2015) stressed that NTL contains various types of information that merit further research [13]. For example, The Defense Meteorological Satellite Program/Operational Linescan System (DMSP/OLS) data has been employed in various applications, including urban extent and extension analysis [14,15], regional economy assessment [16,17], energy releasing events monitoring [18], and fishery research [19]. DMSP/OLS data have significant research value because of the long time period covered. However, these data are too coarse to extract detailed spatial information [20]. Thus, later studiestried to combine DMSP/OLS data with high-resolution remote sensing data to obtain accurate results [21–23]. For example, the Visible Infrared Imaging Radiometer Suite (VIIRS) was launched in 2011 on the Suomi National Polar-Orbiting Partnership (NPP) spacecraft and it provides a new data source with a resolution of 15′ (~500 m), implying a much-improved detail detecting capacity. Shi et al. (2014) employed the VIIRS NTL data to extract built-up urban areas, proving its reliability in urban extent extraction [24]. The combination of VIIRS NTL data with high-resolution remote sensing data has been effective in extracting built-up areas [25,26].The higher spatial resolution of the VIIRS NTL data ensures it anobviously better ability in separating light sources from other land covertypes [27].

The Luojia 1-01 satellite was launched from China on 2 June 2018 and its onboard complementary metal oxide semiconductor can produce high-resolution NTL imagery (130 m). As the world's first professional NTL satellite, Luojia 1-01 has a swath widthof 250 km and it covers the Earth surface every 15 days. Luojia 1-01 data have been used to extract urban extent characteristics [28,29] and investigate artificial light pollution [30]. Compared with previous studies based on other NTL data, such as NPP-VIIRS data, more precise extent of urban impervious surface can be obtained using Luojia 1-01 data Appendix A, Figure A1), due to its superior capacity to detect more details andits wider measurement range [31]. Further, researchers found it feasible to detect urban expansion through the combination of Luojia 1-01 data and other imagery data. The high spatial resolution of the NTL images played a critical role in achieving more accurate resultsin detecting distinct energy-releasingobjects, such as urban impervious surface, population density, or human activity intensity [32,33].

Along with increasing applications of NTL data inresearches, novel methods have been developed to explore specific features of this data source. For example, threshold-based method is widely used to select a specific NTL value to distinguish built-up areas from non-built-up areas [34,35]. With ancillary data such as International Space Station images [32], multiple thresholds can beidentified for extraction in different regions or different time periods [36]. Clustering methods are also commonly applied in urban areas extraction, which is especially useful in large-scale studies [15]. Machine-learning methodsrepresent another active frontier of built-up areas classification with NTL data; related examples include support vector machine [37], artificial neural network [38], and specifically developed methods [39].

In this study, we employed Luojia 1-01 data, VIIRS data, and Landsat 8 data to develop a method for extracting urban built-up areas using kernel density estimation (KDE), taking Nanjing, the capital city of Jiangsu Province of China, as a study area.By comparing the extraction of urban built-up areas using these two NTL datasets and testing the results with the validation data, we intended to answer the following two questions: (1) What is the relative advantages of Luojia 1-01 compared with VIIRS in detecting urban spatial structure?; (2)How dothe searching radius of KDE and the discriminating threshold value affect the effectiveness of KDE in extracting urban built-up areas, especially theurban boundariesand new emerging built-up areas?

## 2. Materials and Methods

### 2.1. Study Region and Data

#### 2.1.1. Remote Sensing Data

Nanjing is a large inland port city, the capital of Jiangsu Province in East China, with a population of 8.436 million and an urban area of 6587 km$^2$ by 2018. Nanjing comprises eleven urban districts (i.e.,Gaochun, Gulou, Lishui, Liuhe, Jiangning, Jianye, Pukou, Qinhuai, Qixia, Xuanwu, Yuhuatai, andthe Jiangbei New District) that are distributed on the south and the north banks of the Yangtze River. The Luojia1-01 data product of Nanjing used in this study was imaged on 23 November 2018. It completely covered the study area with the central geographical coordinates of 117.880537°E/31.883928°N. A Landsat 8 Operational Land Imager (OLI) image on 19 April 2018 was acquired from the Geospatial Data Cloud (http://www.gscloud.cn/, accessed on 10 July 2020). The central coordinates of the image were 118.8335°E/31.7424°N, and the cloud cover was 0.31%. The VIIRS monthly synthetic product acquired in December 2018 for the same region was downloaded from https://www.ngdc.noaa.gov/eog/viirs/download_dnb_composites.html (accessed on 21 November 2019), the website of the National Oceanic and Atmospheric Administration, and included in this study.

The images were clipped to fit the study area. To ensure the accuracy of the area calculations, images were applied using the Albers equal-area conic projection. To reduce the effect of light saturation, aradiometric correction for Luojia1-01 NTL was implemented using the following formula provided by the data distribution website:

$$L = DN^{3/2} \cdot 10^{-10} \tag{1}$$

where *DN* is the digital number representing the image value of each pixel, and *L* represents the corrected radiance of the Luojia1-01 NTL data.

The unit of Luojia 1-01 radiance is W·m$^{-2}$·sr$^{-1}$·μm$^{-1}$, and we converted the unit to nano W·cm$^{-2}$·sr$^{-1}$ which is the unit of VIIRS data. To eliminate georeferencing errors in Luojia 1-01 data, a geometric correction was also done referring to the OpenStreetMap. After the correction, the image matched well with ground objects (Appendix A, Figure A2). For further processing, both NTL images were resampled to the same resolution of Landsat 8 data (30 m) through cubic spline interpolation. Figure 1a,b show the corrected Luojia 1-01 image and the VIIRS image, respectively.

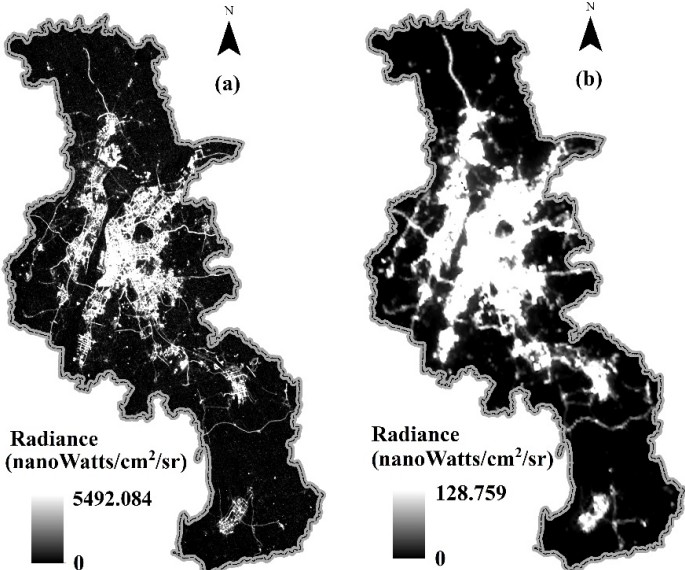

**Figure 1.** City structure of Nanjing City represented by radiometric corrected nighttime light (NTL) images of (**a**) Luojia 1-01 and (**b**) Visible Infrared Imaging Radiometer Suite (VIIRS).

### 2.1.2. Validation Data

Validation data isessential to assess the accuracy of urban built-up areas extracted from the remote sensing data. In our study, the Nanjing Zoning Map and the urban system planning map in the Nanjing Urban Master Plan (Figure 2a,b) were included for validation purpose. The maps were used to evaluate whether the spatial pattern of the extracted built-up area can reflect the actual structure of the urban system.

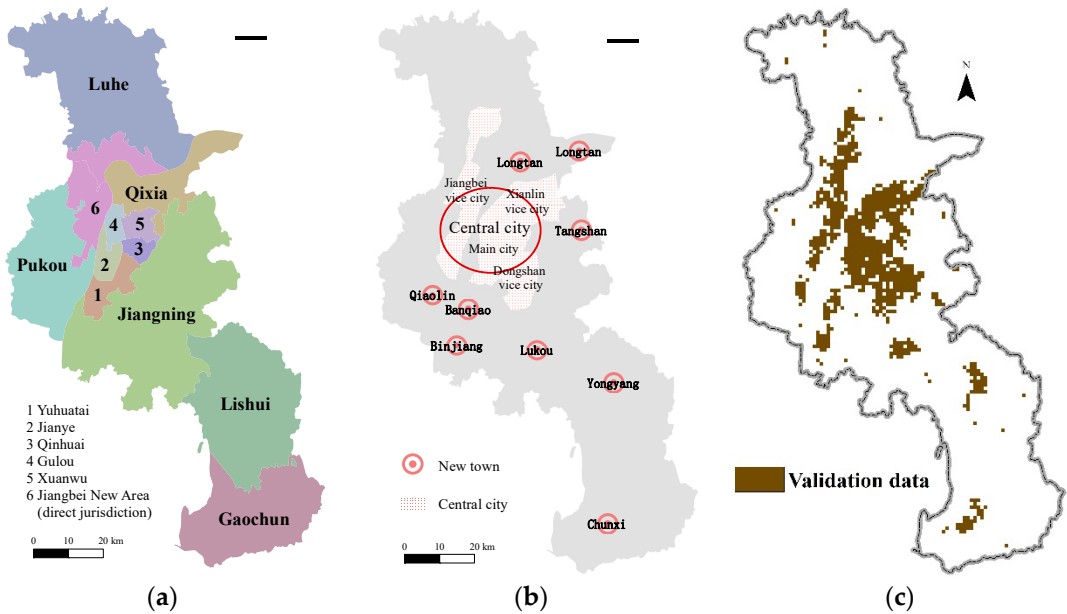

(**a**) (**b**) (**c**)

**Figure 2.** Validation maps of the structure of Nanjing city represented as: (**a**) the zoning mapissued by Jiangsu Provincial Bureau of Surveying Mapping and Geoinformation; (**b**) the urban system map derived from the plan of the Nanjing Jiangbei New District Administrative Committee and the Nanjing Urban Master Plan (2011–2020) issued by Nanjing Municipal Planning and Natural Resources Bureau; (**c**) the built-up area of Nanjing in 2018 from the Resource and Environment Data Cloud Platform.

For accuracy evaluation, we obtained the data of urban built-up areas in 2018 from the Resource and Environment Data Cloud Platform (http://www.resdc.cn/, accessed on 27 January 2020), supported by the Institution of Geographic Sciences and Natural Resources Research, the Chinese Academy of Sciences. This 1-km resolution raster data was derived from the Landsat 8 data through manual visual interpretation (Figure 2c).

### 2.2. Analytical Methods

We used the Vegetation Adjusted NTL Urban Index (VANUI) [40] to extract built-up areas in Nanjing from Luojia 1-01 and VIIRS images. KDE was conducted under different search radii, and then a threshold method was applied to extract high-value pixels as built-up areas. Extraction results were compared with the validation data of urban built-up areas in Nanjing to evaluate the accuracy. A conceptual diagramof methods of this study is shown in Figure 3.

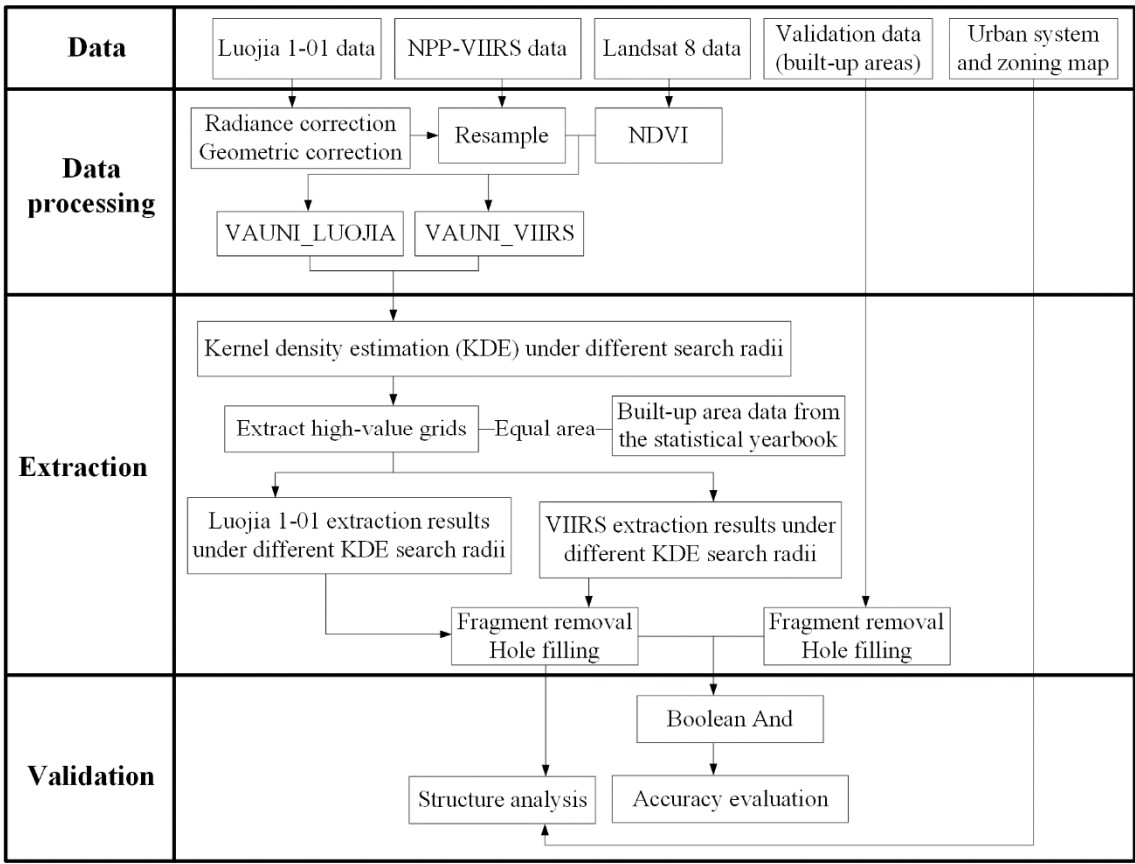

**Figure 3.** The conceptual diagram of analytical procedure of this study.

2.2.1. VANUI for Luojia 1-01 and VIIRS

To improve the sensitivity of light density to the geographical objects it was used to represent [41–43], such as the intensity of economic activities, we combined NTL and Normalized Difference Vegetation Index (*NDVI*) to calculate the VANUI as indices for extracting built-up areas instead of directly using NTL. This spectral index has been proven effective in reducing NTL saturation and increasing variation of data values in core urban areas [40]. The normalized difference vegetation index is an indicator of vegetation coverage:

$$NDVI = \frac{NIR - R}{NIR + R} \tag{2}$$

where *NIR* represents the near-infrared band and *R* represents the red band, i.e., band 5 and band 4 in the Landsat 8 OLI, respectively.

VANUI is defined as Equation (3), where NDVI is derived from Equation (2), and NTL represents the radiance value of Luojia 1-01 and VIIRS data:

$$VANUI = (1 - NDVI) \times NTL \tag{3}$$

*VANUI* derived from Luojia 1-01 and VIIRS datawas respectively calculated (VANUI_LUOJIA and VANUI_VIIRS). Figure 4 shows the spatial structures of Nanjing City derived from VANUI_LUOJIA and VANUI_VIIRS.

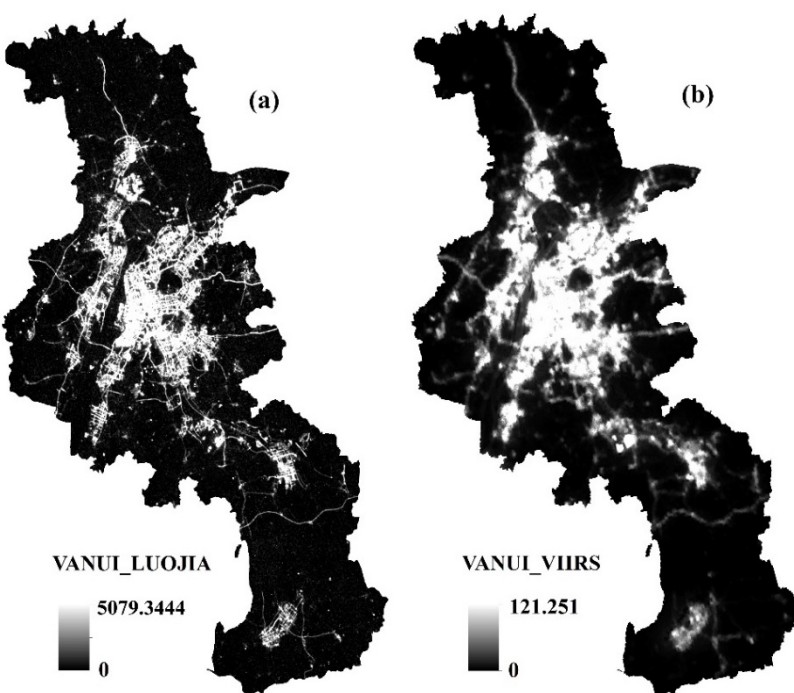

**Figure 4.** Spatial patterns of the urban indices: (**a**) Vegetation Adjusted NTL Urban Index (VANUI)_LUOJIA, derived from Luojia 1-01 data and Normalized Difference Vegetation Index (NDVI); (**b**) VANUI_VIIRS, derived from VIIRS data and NDVI.

### 2.2.2. KDE Method

KDE is a distance-dependent density estimate method, in which the value of each output grid/point represents the accumulative influence of a neighborhood, described by a kernel function, on the focal grid/point density [44]. KDE is generally applied to describe the spatial patterns with lateral overflow, such as the species distribution ranges [45] and road density patterns [46,47]. The density in each output grid cell is calculated by adding the values of all the kernel surfaces where they overlay the grid cell center. KDE is also applicable to NTL data as overflow generally exists in light density. Moreover, the indicative capacity of NTL is variable among geographical objects. Specifically, for urban patterns, the raw data of NTL could underrepresent the blocks with most lights inside turning off at night, such as schools, banks, and parks located in urban areas. Therefore, a smooth of night density is helpful to visualize the urban areas darker at night. Conceptually, a smoothly curved surface is fitted over each point. The surface value is highest at the location of the point and it decreases as the distance increases from the point, until it reaches zero at the bandwidth distance from the point. The bivariate KDE is defined as:

$$\hat{f}(x) = \frac{1}{nh^2} \sum_{i=1}^{n} K\left\{ \frac{x - X_j}{h} \right\} \tag{4}$$

where $n$ is the sample size, $h$ represents the bandwidth, $K$ is the kernel function, the two-dimensional $x$ denotes the vector for which the function is evaluated, and the two-dimensional $X_j$ is the sample vector [48].

The kernel function used here is based on the quartic kernel function as follows:

$$K(x) = \begin{cases} 3\pi^{-1}(1 - x^T x) & x^T x < 1 \\ 0 & otherwise \end{cases} \tag{5}$$

In this study, we integrated the effect of the NTL surrounding each grid point, and classified the output grids with values higher than a specified threshold value as urban built-

up areas. KDE was thus used to compare the spatial patterns of urbanization recognized by the NTL data of Luojia 1-01 and VIIRS.

We converted VANUI_LUOJIA and VANUI_VIIRS into a set of point features at the center of each grid cell. Each point had the same value as the index grid cell from which it was derived. We then used the KDE method to estimate the spatial pattern of the point density to obtain the spatial distribution information for each index. The value of each point was treated as its weight, i.e., the number of calculations for the point. The search radius is an important parameter in the KDE method which has considerable influence on the extraction result. To explore the proper search radius, KDE on VANUI_LUOJIA and VANUI_VIIRS was conducted under a search radius ranging from 100 m to 2000 m with a 100-m interval, resulting in twenty KDE images for each NTL data.

### 2.2.3. Threshold-Based Urban Built-Up Area Extraction

KDE cannot be used to discriminate boundary by itself. Density threshold determining is essential for reasonably deciding the boundaries of urban built-up areas. The most widely employed methods include the mutation detection method [49], empirical threshold method [50], and reference comparison method basing on spatial data [34] or statistical data [51]. In this study, we employed the statistical data to help determine the threshold value in order to identify and extracturban built-up areas. This method has been adopted in previous NTL-based studies to determine the threshold in built-up area extraction and the results proved satisfying [24,52,53].

According to the Nanjing Statistical Yearbook -2019, the built-up area of Nanjingcovered 817 km$^2$ in 2018. We employed this as a benchmark to extract 817 high-value pixels from each KDE image. Then we eliminated individual pixels which were not connected with other extracted pixels, and filled holes to obtain twenty optimum extraction results for Luojia 1-01 and VIIRS data with different searching radii.

### 2.2.4. Extraction Result Evaluation

In this study, the Nanjing zoning boundaries and the Nanjing Urban System Plan map were used to assess the structure of extracted built-up areas, while the built-up area for 2018 was applied to evaluate the extraction accuracy.

As shown in Figure 2c, validation data of the built-up area for 2018 has holes and small fragments of built-up area. Thus, the processes of fragment removal and hole filling in Section 2.2.3 were also carried out before the accuracy evaluation, resulting in a validation data covering 963 km$^2$. Four commonlyused metrics were calculated for accuracy evaluation, including overall accuracy (OA), producer's accuracy (PA), user's accuracy (UA) and Kappa coefficient (KC) [54]. OA presented the ratio of correctly identified pixels, PA indicated the proportion of detected built-up areas in the validation data, UA measured the proportion of truly built-up areas in extraction results, and KC provided an overall assessment of extraction accuracy.

## 3. Results

### 3.1. Extraction Results and the Urban Structure

The KDE-based extraction results using two NTL datawith four searching radii (i.e., 500 m, 1000 m, 1500 m, and 2000 m) were compared separately with the urban structure of Nanjing (Figure 5). In general, built-up areas were mostly concentrated in the central city. Districtsthat had long histories of urban development (e.g., Qinhuai, Xuanwu, and Gulou) were almost fully covered with built-up areas. Regions with rapid economic growth in recent years (such as Jiangbei New District and Jiangning District) also had large built-up areas.

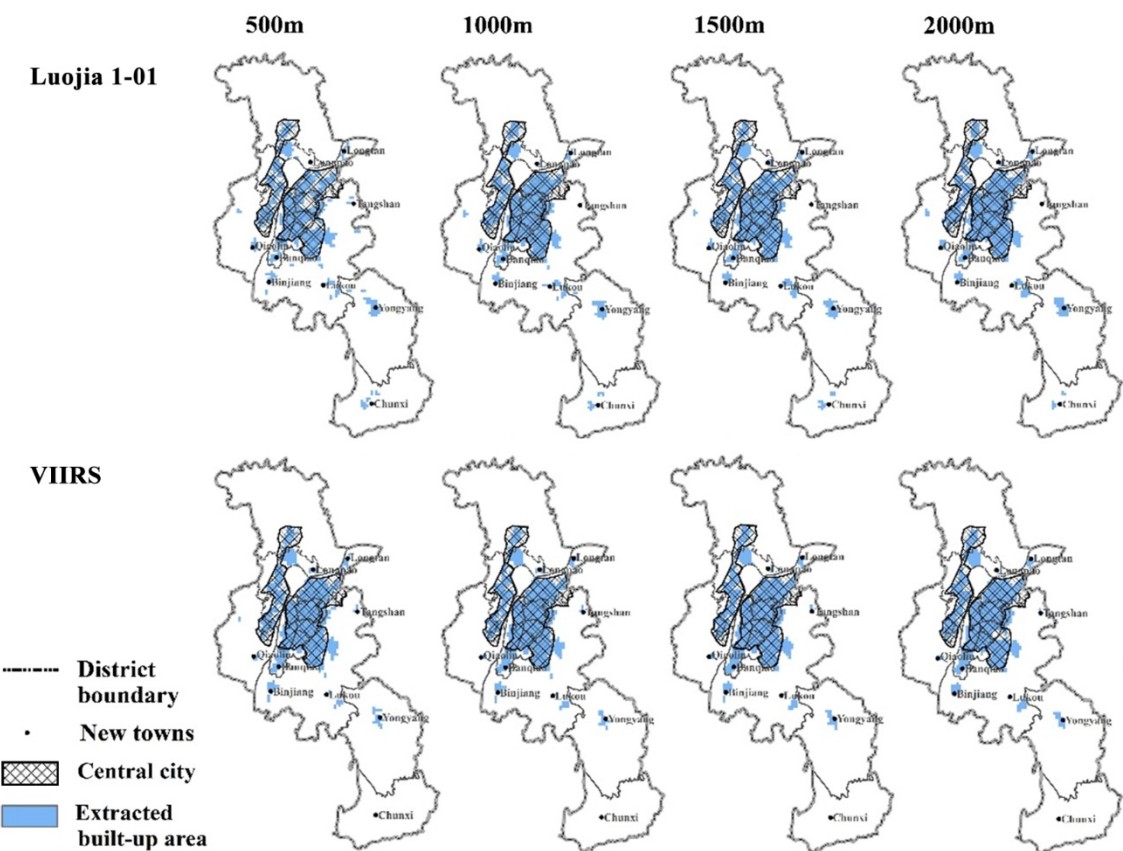

**Figure 5.** Comparison of extraction results and the urban structure of Nanjing, in which blue areas indicate extracted built-up areas, hatched areas indicate the central city, points represent new towns, and boundaries between districts are drawn. Kernel density estimation (KDE) was conducted respectively on VANUI_LUOJIA and VANUI_VIIRS under the search radius of 500 m, 1000 m, 1500 m, and 2000 m.

The urban system of Nanjing is composed of the central city and nine new towns. Under a searching radius of 500 m, Luojia 1-01 identified all nine new towns with the best performance. However, the results didnot agree well with the central city. With the increase of the search radius, the new Tangshan town could not be detected, but the extraction results performed better in the central city. On the other hand, VIIRS was able to detect Tangshan but missed the new Chunxi town in northern Nanjing. In new towns Qiaolin, Lukou, and Yongyang, VIIRS-based extraction results covered less area than that derived from Luojia 1-01 data.

### 3.2. Accuracy Evaluation

Accuracy evaluation of the extraction results using OA, PA, UA, and Kappa metrics were shown in Figure 6, for the raw data (radius as 0 m) of Luojia 1-01, VIIRS, and their KDE images with radius increasing from 100 m to 2000 m; and the confusion matrix at the radius of 0 m (no KDE), 500 m, 1000 m, 1500 m, and 2000 m were selected to show in Table 1. For Luojia 1-01 extractions, the KDE results are more accurate than the raw data when the search radius was larger than 500 m; while for VIIRS, the KDE results were consistently more accurate than its raw data. Overall, the Luojia-based extractions showed higher accuracies than those of the VIIRS-basedresults in the KDEs with the radius over 500 m, but the accuracy of the raw data (resampled to 1 km) of VIIRS was superior to that of Luojia 1-01. The accuracy of Luojia 1-01 extractions showed a unimodal change across the range of search radius and peaked at a radius around 1000 m. ForVIIRS-based extractions, the accuracy increased steadily and peakedat the search radius of about 1600 m.The accuracy of the best extraction result of Luojia 1-01 images (0.937, 0.733, 0.815, and 0.734 for OC, PC,

UC, and Kappa, respectively) was higher than that of VIIRS images (0.935, 0.732, 0.805, and 0.730, correspondingly).

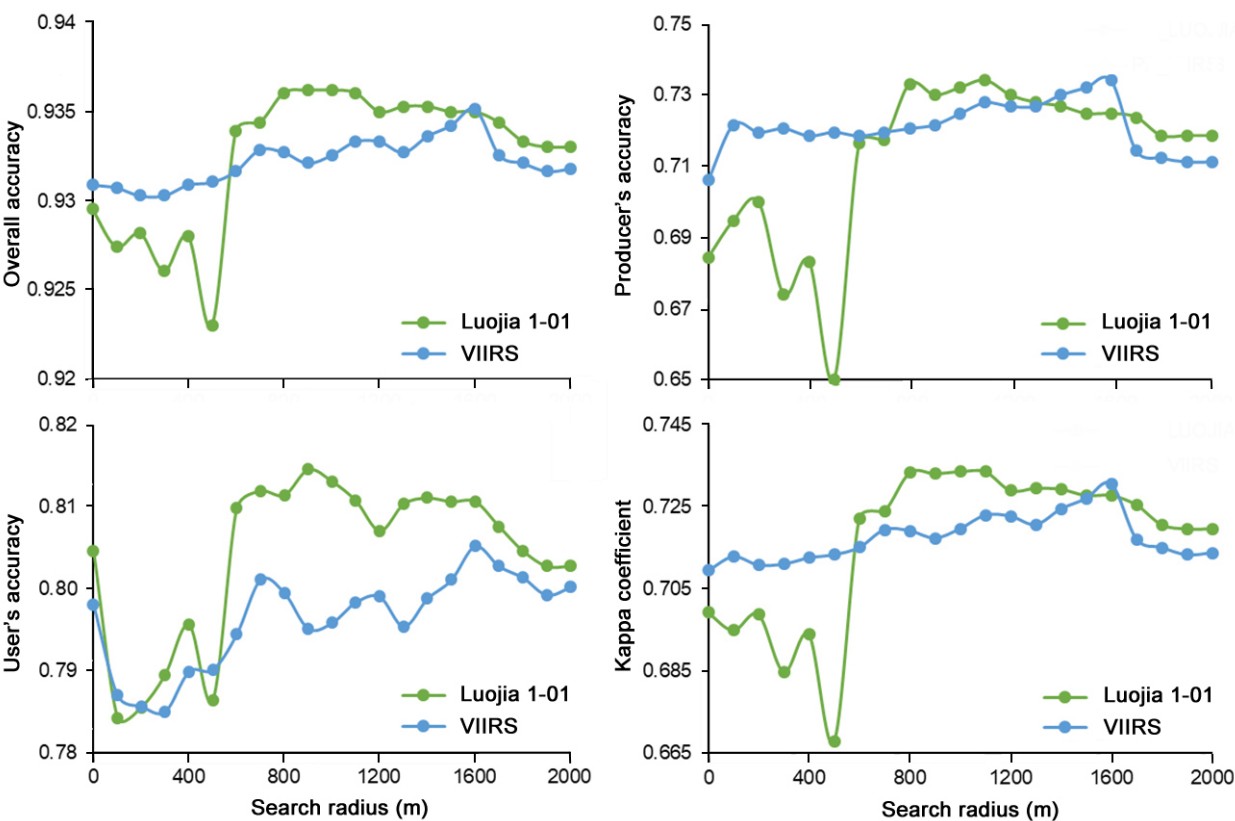

**Figure 6.** The change of accuracy evaluationmetrics for built-up area extractions under different search radii using Luojia 1-01 and VIIRS data products.The evaluation values at the search radius of 0 m represent the raw data of LuoJia 1-01 and VIIRS, respectively, without application of KDE.

**Table 1.** Confusion matrix of extracting built-up area with different search radii using Luojia 1-01 and VIIRS data products. BA: built-up areas; NA: non-built-up areas.

| Search Radius | | Raw (0 m) | | 500 m | | 1000 m | | 1500 m | | 2000 m | |
|---|---|---|---|---|---|---|---|---|---|---|---|
| | | BA | NA | BA | NA | BA | NA | BA | NA | BA | NA |
| Luojia 1-01 | BA | 10.0% | 4.6% | 9.5% | 2.6% | 10.7% | 2.5% | 10.6% | 2.5% | 10.5% | 2.6% |
| | NA | 2.4% | 83.0% | 5.1% | 82.8% | 3.9% | 82.9% | 4.0% | 82.9% | 4.1% | 82.8% |
| VIIRS | BA | 10.3% | 4.3% | 10.5% | 2.8% | 10.6% | 2.7% | 10.7% | 2.7% | 10.4% | 2.6% |
| | NA | 2.6% | 82.8% | 4.1% | 82.6% | 4.0% | 82.7% | 3.9% | 82.7% | 4.2% | 82.8% |

As for the spatial distribution of extraction results (Figure 7), difference can be found between the KDE patterns of two NTL datasets, as well as among different search radii. VIIRS showed relatively poor results in the southern suburbs and failed to detect the most southern built-up area. Under the search radius of 500 m, the KDE of Luojia 1-01 was not able to fully extract the central part of built-up areas, whereas under the search radius of 1000 m, the extraction matched best with the validation data in the central part. However, as the radius continued to increase, omission errors could be found near the margins. Similarly, omission of built-up areas revealed in VIIRS-based extraction when the search radius was 2000 m.

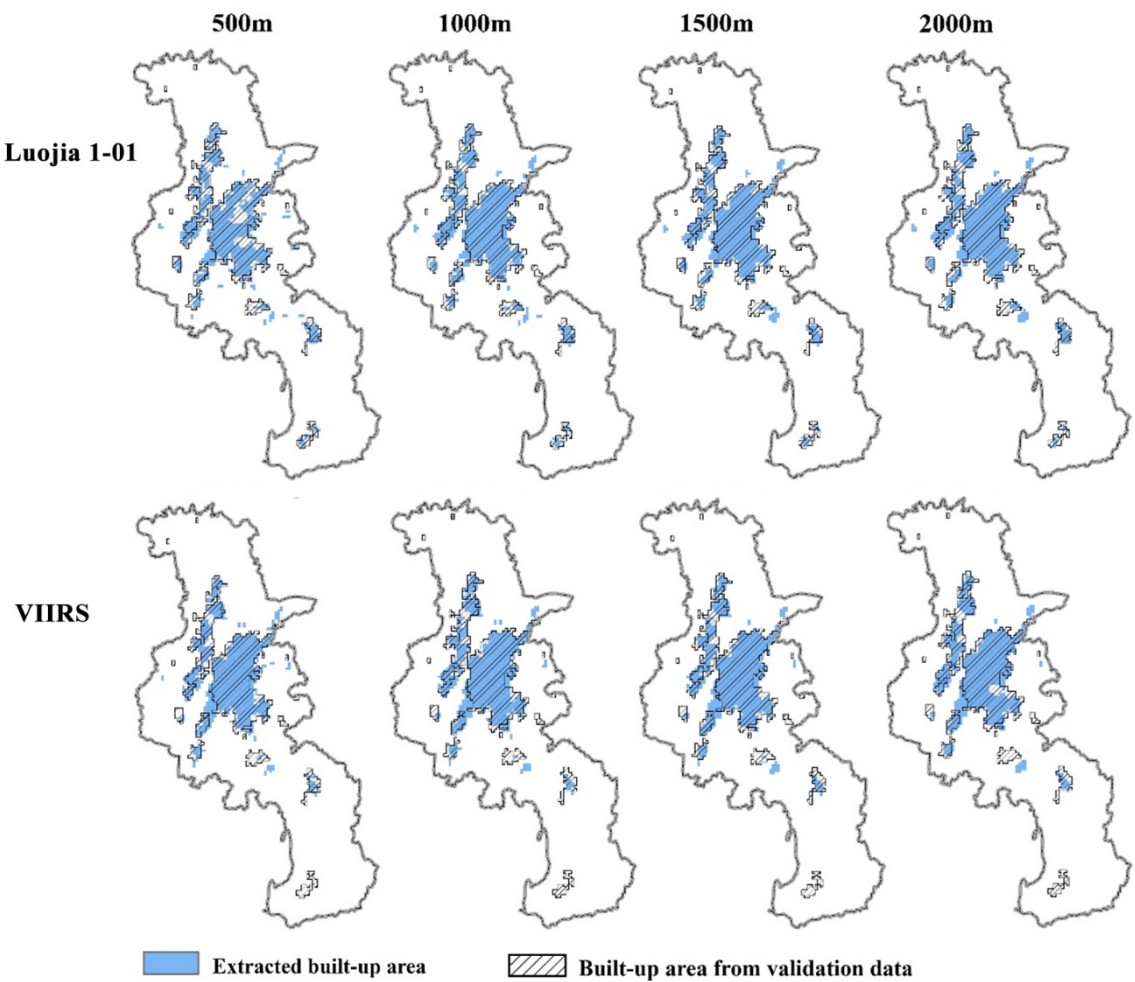

**Figure 7.** Comparison of extraction results and the validation data for urban built-up areas, in which blue areas indicate extracted built-up areas and hatched areas indicate built-up areas from the validation data. KDE was conducted respectively on VANUI_LUOJIA and VANUI_VIIRS under the search radius of 500 m, 1000 m, 1500 m, and 2000 m.

## 4. Discussion

Being recognized as a useful indicator of human activity intensity, NTL data have been increasingly applied for urban structure analyses [55,56]. The technical features of the first professional NTL satellite Luojia 1-01 have been reported earlier [57–59]. After comparing the built-up area extraction results obtained using different researching radii, we found that the threshold-constrained KDEs for both Luojia 1-01 and VIIRS dataeffectively extracted the built-up areas in the urban center and the false extraction of water bodies (such as the Yangtze River and Xuanwu Lake) was satisfactorily avoided. However, there were substantial differences in detection of urban built-up area boundaries, especially in the suburbs. In addition, the search radius of KDE had considerable effects on extraction results, and differed between the two NTL data sources.

The effectiveness of extracting the urban built-up area boundaries differed between the Luojia 1-01 and VIIRS datasets. In the central city, the NTL data of Luojia 1-01 and VIIRS produced comparatively consistent results. However, Luojia 1-01 was better at identifying new growing urban cores, showing the advantage of Luojia 1-01 data with a finer resolution in capturing the emergent urban structure under a proper KDE search radius. The higher spatial resolution of Luojia 1-01 images provide a lesserdegree of mixture of land use types and light sources, and warrant it a higher sensitivity to the change of NTL environment than the coarser-resolution VIIRS images. In particular, moreomission errors occurred in VIIRS-based extraction in the suburbs, where the urban built-up areas

were more fragmented and mixed with other surrounding land-use types. Nevertheless, the daily revisit interval of VIIRS data warrant it a much higher sensitivity to the dynamics of ground processes, although the 15-day temporal resolution of Luojia 1-01 data generally satisfies the requirement for urban built-up area analysis. It is also importantto keep in mind that the difference in local acquisition time for VIIRS (01:30 a.m.) and Luojia 1-01 (10:30 p.m.) could also cause difference in their extraction results.

The KDE-based extraction of urban built-up areas has several advantages compared with direct extraction from raw NTL data. First of all, as a grid data of on-site light intensity, NTL is a useful indicator of nighttime human activity intensity. However, one type of NTL data has a fixed spatial resolution. When it is used to detect the pattern of a particular energy-releasing spatial process such as urban development, a high-resolution NTL data could miss the urban buildings or blocks with most lights turning off at night, such as those of banks, schools, museums, and administrative offices, as well as city parks. These areas are generally scattered in the center of urban areas, but normally empty and darker at night, compared with the areas of active nightlife, such as business and entertainment centers. Therefore, the new high-resolution NTL data may generate more bias in detecting urban areas compared with old coarser data, as demonstrated by the lower accuracy values of Luojia 1-01 than VIIRS for their raw data and short search radius ($\leq$500 m) KEDs (Figure 6). This flaw of high-resolution NTL data can be offset by applying KDE that makes use of the spatial autocorrelation of NTLs with a proper search radius adaptive to the actual lamination environment, rather than being limited by the fixed scale of NTL data itself. Actually, the accuracy of KDE with the optimal search radius was obviously higher that the raw image for both NTL data, and the best KDE of Luojia 1-01 was more accurate than that of VIIRS. Our application of KDE confirmed it as a useful method of urban detecting using high-resolution NTL data, whichsmooth the NTL space by integrating the surrounding NTL values on the focal pixels, and thus reasonably "erase" the darker points within a continuous urban area. Secondly, in a KDE image, high-value grids have high accumulative light density within the search radius. Unlike the raw data, this accumulation can help to identify "NTL hotspots" with a threshold light valueand spatial magnitude that indicate an emergent urban core. Moreover, KDE can be more reasonable and accurate in detecting the urban boundary, out of which the NTL density drops below the threshold value that is determined by the accumulation of surrounding light intensity within a radius, rather than that at a single point.This effect is especially helpful for NTL data of upcoming higher spatial resolution. Urban built-up areas comprise centralized and contiguous areas, and the extraction of their boundaries is traditionally based on manual interpretation or automatic identification according to the edge point density [6]. The KDE method integrated the effects of the surrounding pixels on each output grid, which excluded small or narrowly illuminated areas and helped to extract "centralized and contiguous" urban built-up areas. Our results indicated that, regardless of whether the Luojia 1-01 NTL or VIIRS NTL was employed, the KDE-based extraction results could map the built-up urban areas precisely. This method should be particularly useful when the core urban areas have to be highlighted and the emerging new towns need to be identified, as in the cases of urban planning.

In the KDE process, the search radius is an important parameter which can largely influence the extraction result. As shown in Figure 6, both large and small radii resulted in omission of built-up areas. When the search radius is too small, the KDE process only includes the influence of a narrow neighborhood. Thus, some lighted areas with dark intervals could be left out of the extraction result. If the search radius is too large, built-up areas near the boundaries may be integrated into the non-built-up area, or some disconnected built-up areas will be misconnected. The proper search radius should be the point when the extracted built-up area decreases with the increase of the radius. For Luojia 1-01 and VIIRS datasets used in our case, proper search radii appeared to be similar (1000 m and 1600 m, respectively), indicatingthat the KDE optimal radius might not be optimal per see, but subject to the spatial resolution of the NTL data, the validation data, and the scale of fragmentation feature of the geographical objects.

For both Luojia 1-01 NTL data and the KDE method, there remains much to be explored in future studies. There are other factors thatcan affect the data of Luojia 1-01 images, as demonstrated in other NTL data sources, and thus may also impact the extraction result that requires further work to estimate;these include the seasonal changes in nighttime light brightness [60] and diurnal change of satellite overpass time [61], as well as the effect of satellite observation angle on the nighttime light [62].The radiometric correction formula for Luojia1-01 was obtained after running smoothly for several months. This data source will be more reliable after further corrections aremade for different regions and time domains, and thus expected to provide long-term and high-resolution NTL information to support various applications in research, management, and policy-making; for example, the possible evaluation of new request for energy in newly detected urban areas, as well as the trend of $CO_2$ level in the atmosphere. Luojia 1-01 images from June to November 2018 can be downloaded at present. With more images available in the future, the performance of Luojia 1-01 in detecting urban area changes across different temporal resolution levels could be discussed. Given the effectiveness in the KDE extraction results, choosinga proper search radius would becritical fora successful application of this method. Our study shows the effectiveness of the KDE method in a high-luminosity area; the universal principle to select the optimal search radius in variable luminosity contexts remains for further exploration.

## 5. Conclusions

The comparison with VIIRS indicates that the first professional NTL satellite Luojia 1-01 provides a reliable new data resource of nighttime light remote sensing. Its substantially improved spatial resolution is more sensitive to nighttime light variation, and thus benefits the accurate extraction of the spatial structure of urban built-up areas, especially the urban boundary andthe new growing urban cores. The application of KDE combinedwith a properly determined threshold can be used to make use of the spatial autocorrelation in NTL information.This improvement would be critical for the capture capacity of upcoming higher-resolution NTL data inapplications of overall spatial pattern detection; and a proper searching radius and NTL threshold value is critical for an optimized KDE result. The high agreement between the extraction result and the validation dataindicates the potentialof Luojia 1-01 data in widespread applicationsincluding urban study and planning.

**Author Contributions:** Conceptualization, Z.S. and Y.W.; methodology, Y.W. and Z.S.; formal analysis, Y.W.; writing—original draft preparation, Y.W.; writing—review and editing, Z.S.; funding acquisition, Z.S. All authors have read and agreed to the published version of the manuscript.

**Funding:** This research was funded by the Key Research and Development Plan of the Ministry of Science and Technology of China, grant number 2017YFC0505200, and the project of National Natural Science Foundation of China, grant number 41371190.

**Institutional Review Board Statement:** Not applicable.

**Informed Consent Statement:** Informed consent was obtained from all subjects involved in the study.

**Acknowledgments:** This study is sponsored by the Key Research and Development Plan of the Ministry of Science and Technology of China [2017YFC0505200], and the project of National Natural Science Foundation of China [41371190]. We are grateful for the helpful comments and suggestions from Fang Qiu from University of Texas at Dallas.

**Conflicts of Interest:** The authors declare no conflict of interest. The funders had no role in the design of the study; in the collection, analyses, or interpretation of data; in the writing of the manuscript, or in the decision to publish the results.

## Appendix A

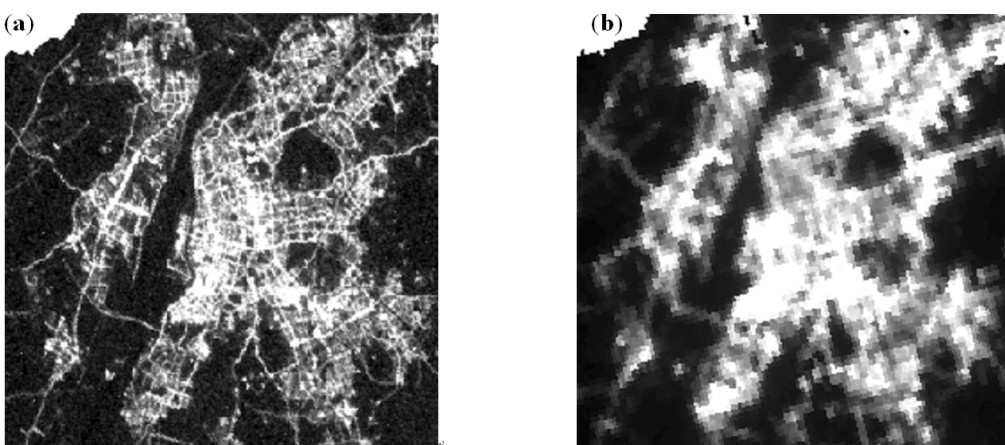

**Figure A1.** Comparison of the spatial resolutions of nighttime light (NTL) remote sensing data sources in the study area. (**a**) Luojia 1-01 image acquired on 23 November 2018. (**b**) VIIRS monthly synthetic product for December 2018.

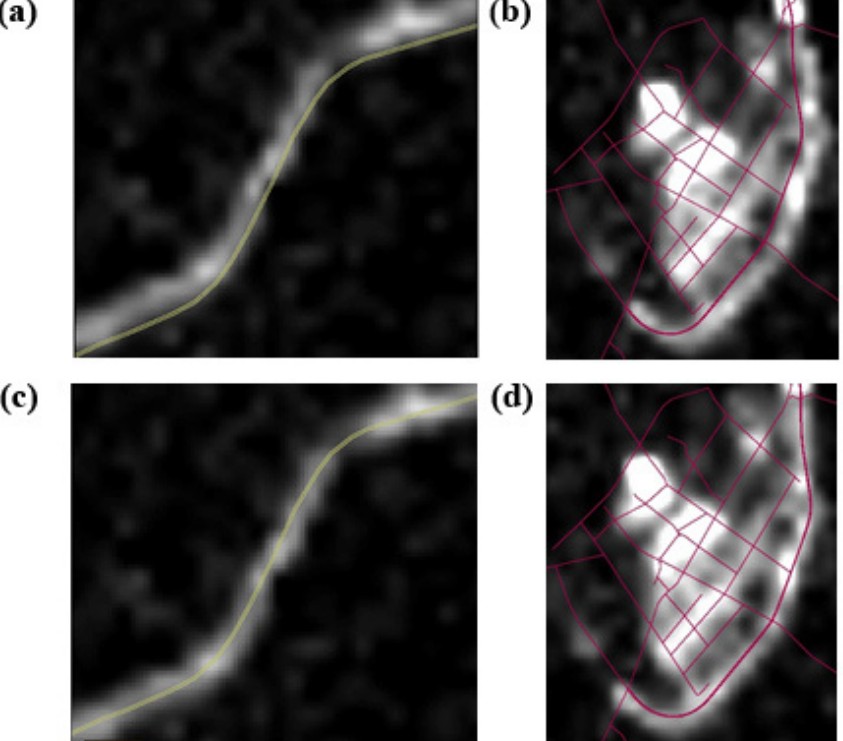

**Figure A2.** Luojia 1-01 image before (**a**,**b**) and after (**c**,**d**) the geometric correction.

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
