# Peer review of "Comparing Luojia 1-01 and VIIRS Nighttime Light Data in Detecting Urban Spatial Structure Using a Threshold-Based Kernel Density Estimation"

_remotesensing, doi:10.3390/rs13081574_

Round 1
Reviewer 1 Report
I suggest to define the possible use of the evaluation of the proposed approach to detect the urban spatial structure (this aspect is absented in the paper).
For instance, the possible evaluation of the new request of water or energy in the new spatial structure identified, taking into account the increasing number of final users
Or the different environmental aspects correlated to the pollution or value of CO2 in the air when the population grows
Author Response
Thank you for your helpful suggestions. In the second to last paragraph of Discussion, we mentioned some possible applications of Luojia for environmental assessments, which may address your concerns.
Reviewer 2 Report
Review of: “Comparing Luojia 1-01 and VIIRS nighttime light data in detecting urban spatial structure using a threshold-based kernel 3 density estimation”
This study compares the VIIRS NTL data with the more recent Luoija 1-01 data, to highlight the variations across the two systems as they are being specifically applied to measuring the extent of urban development from space. There is value in this comparison, for certain, but there are some general issues with the manuscript that – I think – would benefit from further clarification.
- Temporal accuracy: the authors focus primarily on how each system compares with respect to identifying the extent of urbanization *spatially*. I am curious, however, to know if there are some differences between the systems with respect to variations *over time*. In particular, there are three relevant comparisons: (i) What is the smallest temporal unit (month, year, week) by which we can observably detect changes in urban area, (ii) which system is better at detecting these changes, and (iii) how is the performance of each system varies across different temporal resolution levels, similarly to how it varies across geospatial resolution level levels. I understand this might take a lot of work, and if so, I don’t expect the authors conduct = all these exercises. But it would be helpful to at least highlight and/or discuss these temporal issues in some fashion.
- The authors use actual data from statistical information books to decide on the number of pixels used to validate performance (pg. 5). I wonder if the authors could discuss a bit more performance validation in areas where such information is unavailable or where overall luminosity is relatively low (especially where urbanization does not consist of solely, or even mostly, constructed areas, such as in sub-Saharan Africa).
- On a related note, the authors focus on Nanking, which is a high-luminosity area. If one moves to low luminosity regions such as central Asia or sub-Saharan Africa, would Luoija 1-01 still perform better than VIIRS at low resolution levels?
- General framing: when I started reading the paper and until I pretty much reached to the Discussion section, I thought the focus was on comparing a new system (Luoija 1-01) to an older one (VIIRS). However, the discussion section is heavily focused on motivating the KDE-based approach. So, in a way, the paper reads as two separate studies: one that makes said comparison, and one that motivates the KDE-based approach. I would recommend that the authors either make this dual-focus clear from the start, or else highlight the comparison, possibly discussing the KDE-based method in a separate note.
- Additionally, the authors mention throughout the text that NTL is used in a wide variety of substantive applications and conclude their discussion section stating there are key implications for research and policy. Yet, the discussion section is focused more on particular aspects related to the method and the inter-system comparison, rather than to explicating these implications. I would recommend that the discussion section will be rewritten in a way that gives more centrality to the general application of the new data and/or new method (pending on the authors’ response to point 4) across different relevant research fields.
Author Response
1. Temporal accuracy: the authors focus primarily on how each system compares with respect to identifying the extent of urbanization *spatially*. I am curious, however, to know if there are some differences between the systems with respect to variations *over time*. In particular, there are three relevant comparisons: (i) What is the smallest temporal unit (month, year, week) by which we can observably detect changes in urban area, (ii) which system is better at detecting these changes, and (iii) how is the performance of each system varies across different temporal resolution levels, similarly to how it varies across geospatial resolution level levels. I understand this might take a lot of work, and if so, I don’t expect the authors conduct all these exercises. But it would be helpful to at least highlight and/or discuss these temporal issues in some fashion.
Reply: Thank you for your valuable comments. The revisit interval of Luojia 1-01 is 15 days, much longer than the daily interval of VIIRS-DNB. Therefore, VIIRS is obviously superior in detecting rapid dynamics and spatial processes on the ground; although a 15 days revisit interval is adequate for exploring the spatial structure change of urban areas. We added this point at the end of the 2nd paragraph of Discussion. However, since this study specifically focuses on the application of KDE for two NLT data sources in detecting the spatial pattern of the urban built-up area, we did not expand our discussion on the temporal aspect.
2. The authors use actual data from statistical information books to decide on the number of pixels used to validate performance (pg. 5). I wonder if the authors could discuss a bit more performance validation in areas where such information is unavailable or where overall luminosity is relatively low (especially where urbanization does not consist of solely, or even mostly, constructed areas, such as in sub-Saharan Africa).
Reply: Thank you for your helpful comments. Our study presented an application of KDE on Luojia 1-01 and VIIRS data in extracting continuous built-up areas, which represents the spatial structure of the city. The performance of this application was well validated by the independent statistical data. In other regions where the statistical data is unavailable, other remote sensing data such as Landsat can be used for NLT data validation, as practiced in many published studies. For where urbanization does not consist of solely constructed areas, the optimal search radius and threshold may be better adjusted smaller to ensure satisfactory results. Although this is not the main focus of our research, the optimal search radius for KDE under different environment of luminosity deserves further research. We mentioned this point at the end of Discussion.
3. On a related note, the authors focus on Nanking, which is a high-luminosity area. If one moves to low luminosity regions such as central Asia or sub-Saharan Africa, would Luoija 1-01 still perform better than VIIRS at low resolution levels?
Reply: Thank you for your comments. An earlier study compared the feeble NTL detecting ability of Luojia 1-01, DMSP/OLS and VIIRS in a set of light intensity environments from cities, counties, towns to villages (Liu et al. 2019). They found LJ1‐01 was more suitable for detection feeble NTL objects, as supported by the evidences: (1) In the study area, a suitable noise cutoff threshold of LJ1‐01 image can be set to 0.1 nano‐Wcm−2sr−1, which is lower than that of VIIRS image (0.3 nano‐Wcm−2sr−1), and this enables LJ1‐01 to reserve more information of NTL, especially the feeble NTL. (2) the minimum area that can be identified by NTL footprints from LJ1‐01 is 0.02 km2, while that of VIIRS and DMSP are 0.3 km2 and 4.5 km2, respectively.
Reference:
Li X., Liu Z., Chen X., Sun J. 2019. Assessing the ability of Luojia 1‐01 imagery to detect feeble nighttime lights. Sensor, 19, 3708; doi:10.3390/s19173708
4. General framing: when I started reading the paper and until I pretty much reached to the Discussion section, I thought the focus was on comparing a new system (Luoija 1-01) to an older one (VIIRS). However, the discussion section is heavily focused on motivating the KDE-based approach. So, in a way, the paper reads as two separate studies: one that makes said comparison, and one that motivates the KDE-based approach. I would recommend that the authors either make this dual-focus clear from the start, or else highlight the comparison, possibly discussing the KDE-based method in a separate note.
Reply: Thank you for your helpful suggestion. At the end of introduction section, we clarified our dual-focus that “we intended to answer following two questions: 1) What is the relative advantages of Luojia 1-01 compared with VIIRS in detecting urban spatial structure? 2) How do the searching radius of KDE and the discriminating threshold value affect the effectiveness of KDE in extracting urban built-up areas, especially the urban boundaries and new emerging built-up areas?”. In the revised manuscript, we made the two focuses clearer by rewording the abstract (Lines18-26), and discussion part.
5. Additionally, the authors mention throughout the text that NTL is used in a wide variety of substantive applications and conclude their discussion section stating there are key implications for research and policy. Yet, the discussion section is focused more on particular aspects related to the method and the inter-system comparison, rather than to explicating these implications. I would recommend that the discussion section will be rewritten in a way that gives more centrality to the general application of the new data and/or new method (pending on the authors’ response to point 4) across different relevant research fields.
Reply: Thank you for your helpful comment. Actually, there have been many papers addressing the application fields of Luojia 1-01 data (e.g. Li et al., 2020; Wang et al., 2020; Zhang et al., 2020), and of course much more on that of the VIIRS data. Thus we intend to focus on the reliability of Luojia 1-01 in detecting spatial structure of urban areas, with the help of the KDE method. Thus, we rewrote the discussion section to make our dual-focus clearer.
Reference:
Li, C., Yang, W., Tang, Q., Tang, X., Lei, J., Wu, M., & Qiu, S. 2020. Detection of multi- dimensional poverty using Luojia 1-01 nighttime light imagery. Journal of the Indian Society of Remote Sensing, 48, 963–977.
Wang L., Fan H., Wang Y. 2020. Improving population mapping using Luojia 1-01 nighttime light image and location-based social media data. Science of the Total Environment, 730: 139148
Zhang, C., Pei, Y., Li, J., Qin, Q., and Yue J. 2020, Application of Luojia 1-01 Nighttime Images for Detecting the Light Changes for the 2019 Spring Festival in Western Cities, China. Remote Sensing 12(9), 1416; doi.org/10.3390/rs12091416
Reviewer 3 Report
I suggest publication of this paper in present form.
Author Response
Reply: Thank you for your positive comment on our manuscript.
Round 2
Reviewer 2 Report
I am satisfied with the authors' revisions.
Author Response
- Temporal accuracy: the authors focus primarily on how each system compares with respect to identifying the extent of urbanization *spatially*. I am curious, however, to know if there are some differences between the systems with respect to variations *over time*. In particular, there are three relevant comparisons: (i) What is the smallest temporal unit (month, year, week) by which we can observably detect changes in urban area, (ii) which system is better at detecting these changes, and (iii) how is the performance of each system varies across different temporal resolution levels, similarly to how it varies across geospatial resolution level levels. I understand this might take a lot of work, and if so, I don’t expect the authors conduct all these exercises. But it would be helpful to at least highlight and/or discuss these temporal issues in some fashion.
Reply: Thank you for your valuable comments. The revisit interval of Luojia 1-01 is 15 days, much longer than the daily interval of VIIRS-DNB. Therefore, VIIRS is obviously superior in detecting rapid dynamics and spatial processes on the ground; although a 15 days revisit interval is adequate for exploring the spatial structure change of urban areas. We added this point at the end of the 2nd paragraph of Discussion. However, since this study specifically focuses on the application of KDE for two NLT data sources in detecting the spatial pattern of the urban built-up area, we did not expand our discussion on the temporal aspect.
- The authors use actual data from statistical information books to decide on the number of pixels used to validate performance (pg. 5). I wonder if the authors could discuss a bit more performance validation in areas where such information is unavailable or where overall luminosity is relatively low (especially where urbanization does not consist of solely, or even mostly, constructed areas, such as in sub-Saharan Africa).
Reply: Thank you for your helpful comments. Our study presented an application of KDE on Luojia 1-01 and VIIRS data in extracting continuous built-up areas, which represents the spatial structure of the city. The performance of this application was well validated by the independent statistical data. In other regions where the statistical data is unavailable, other remote sensing data such as Landsat can be used for NLT data validation, as practiced in many published studies. For where urbanization does not consist of solely constructed areas, the optimal search radius and threshold may be better adjusted smaller to ensure satisfactory results. Although this is not the main focus of our research, the optimal search radius for KDE under different environment of luminosity deserves further research. We mentioned this point at the end of Discussion.
- On a related note, the authors focus on Nanking, which is a high-luminosity area. If one moves to low luminosity regions such as central Asia or sub-Saharan Africa, would Luoija 1-01 still perform better than VIIRS at low resolution levels?
Reply: Thank you for your comments. An earlier study compared the feeble NTL detecting ability of Luojia 1-01, DMSP/OLS and VIIRS in a set of light intensity environments from cities, counties, towns to villages (Liu et al. 2019). They found LJ1‐01 was more suitable for detection feeble NTL objects, as supported by the evidences: (1) In the study area, a suitable noise cutoff threshold of LJ1‐01 image can be set to 0.1 nano‐Wcm−2sr−1, which is lower than that of VIIRS image (0.3 nano‐Wcm−2sr−1), and this enables LJ1‐01 to reserve more information of NTL, especially the feeble NTL. (2) the minimum area that can be identified by NTL footprints from LJ1‐01 is 0.02 km2, while that of VIIRS and DMSP are 0.3 km2 and 4.5 km2, respectively.
Reference:
Li X., Liu Z., Chen X., Sun J. 2019. Assessing the ability of Luojia 1‐01 imagery to detect feeble nighttime lights. Sensor, 19, 3708; doi:10.3390/s19173708
- General framing: when I started reading the paper and until I pretty much reached to the Discussion section, I thought the focus was on comparing a new system (Luoija 1-01) to an older one (VIIRS). However, the discussion section is heavily focused on motivating the KDE-based approach. So, in a way, the paper reads as two separate studies: one that makes said comparison, and one that motivates the KDE-based approach. I would recommend that the authors either make this dual-focus clear from the start, or else highlight the comparison, possibly discussing the KDE-based method in a separate note.
Reply: Thank you for your helpful suggestion. At the end of introduction section, we clarified our dual-focus that “we intended to answer following two questions: 1) What is the relative advantages of Luojia 1-01 compared with VIIRS in detecting urban spatial structure? 2) How do the searching radius of KDE and the discriminating threshold value affect the effectiveness of KDE in extracting urban built-up areas, especially the urban boundaries and new emerging built-up areas?”. In the revised manuscript, we made the two focuses clearer by rewording the abstract (Lines18-26), and discussion part.
- Additionally, the authors mention throughout the text that NTL is used in a wide variety of substantive applications and conclude their discussion section stating there are key implications for research and policy. Yet, the discussion section is focused more on particular aspects related to the method and the inter-system comparison, rather than to explicating these implications. I would recommend that the discussion section will be rewritten in a way that gives more centrality to the general application of the new data and/or new method (pending on the authors’ response to point 4) across different relevant research fields.
Reply: Thank you for your helpful comment. Actually, there have been many papers addressing the application fields of Luojia 1-01 data (e.g. Li et al., 2020; Wang et al., 2020; Zhang et al., 2020), and of course much more on that of the VIIRS data. Thus we intend to focus on the reliability of Luojia 1-01 in detecting spatial structure of urban areas, with the help of the KDE method. Thus, we rewrote the discussion section to make our dual-focus clearer.
Reference:
Li, C., Yang, W., Tang, Q., Tang, X., Lei, J., Wu, M., & Qiu, S. 2020. Detection of multi- dimensional poverty using Luojia 1-01 nighttime light imagery. Journal of the Indian Society of Remote Sensing, 48, 963–977.
Wang L., Fan H., Wang Y. 2020. Improving population mapping using Luojia 1-01 nighttime light image and location-based social media data. Science of the Total Environment, 730: 139148
Zhang, C., Pei, Y., Li, J., Qin, Q., and Yue J. 2020, Application of Luojia 1-01 Nighttime Images for Detecting the Light Changes for the 2019 Spring Festival in Western Cities, China. Remote Sensing 12(9), 1416; doi.org/10.3390/rs12091416
